# Novel Liquid Chitosan-Based Biocoagulant for Treatment Optimization of Fish Processing Wastewater from a Moroccan Plant

**DOI:** 10.3390/ma14237133

**Published:** 2021-11-23

**Authors:** Nisrine Nouj, Naima Hafid, Noureddine El Alem, Igor Cretescu

**Affiliations:** 1Material and Environmental Laboratory, Department of Chemistry, Faculty of Sciences of Agadir, IBN ZOHR University, Agadir 80000, Morocco; hafidnaima@yahoo.fr (N.H.); n.elalem@uiz.ac.ma (N.E.A.); 2Department of Environmental Engineering and Management, Faculty of Chemical Engineering and Environmental Protection, “Gheorghe Asachi” Technical University of Iasi, 700050 Iasi, Romania

**Keywords:** biocoagulation, seafood processing wastes, fish by-products, industrial effluents, response surface methodology, Box–Behnken experimental design

## Abstract

A novel liquid chitosan-based biocoagulant for treating wastewater from a Moroccan fish processing plant was successfully prepared from shrimp shells (*Parapenaeus longirostris*), the most abundant fish by-products in the country. The shells were characterized using scanning electron microscopy, energy-dispersive X-ray spectroscopy, X-ray diffraction, and Fourier transforms infrared spectroscopy. Using chitosan without adding acetic acid helps to minimize its negative impact on the environment. At the same time, the recovery of marine shellfish represents a promising solution for the management of solid fish waste. In order to test the treatment efficiency of the biocoagulant developed, a qualitative characterization of these effluents was carried out beforehand. The optimization process was conducted in two steps: jar-test experiments and modeling of the experimental results. The first step covered the preliminary assessment to identify the most influential operational parameters (experimental conditions), whereas the second step concerned the study of the effects of three significant operational parameters and their interactions using a Box–Behnken experimental design. The variables involved were the concentration of coagulant (X_1_), the initial pH (X_2_), and the temperature (X_3_) of the wastewater samples, while the responses were the removal rates of turbidity (Y_1_) and BOD_5_ (Y_2_). The regression models and response surface contour plots revealed that chitosan as a liquid biocoagulant was effective in removing turbidity (98%) and BOD_5_ (53%) during the treatment. The optimal experimental conditions were found to be an alkaline media (pH = 10.5) and a biocoagulant dose of 5.5 mL in 0.5 L of fish processing wastewater maintained at 20 °C.

## 1. Introduction

Wastewater effluents from fish processing plants (fish processing wastewater, FPW) present an environmental issue because of complications in their treatment due to the different components presented in the effluents. This sector generates USD 207,407,800.00 per year on export [1]. In Morocco, the subject of this study, the effluent volume is not only increasing, but the fish processing industry is also responsible for a large quantity of fish-related solid wastes, including bones, skins, scales, fins, and swim bladders, constituting 36% (and maybe representing up to 60%) of the raw material mass [2,3]. These solid wastes could be potential sources of income if sufficient assessment is carried out. At the same time, the production units generate wastewater that harms the environment, and this must be treated before discharge. As a result, the rejected by-products can be involved (by using the proposed methodology) in the preparation of products for wastewater treatment.

Different processes were carried out to treat this wastewater in order to protect the environment. Among the possible treatments, there is coagulation–flocculation; the most widely used process, and which addresses the need to replace chemical coagulants with natural ones to preserve the environment. In order to find a natural coagulant, chitosan was used in many studies and for many applications [4,5]; including its extraction from the most abundant fish by-products: shrimp shells, especially *Parapenaeus longirostris* [6]. Valuing this waste for its contents is a solution to reduce fish solid waste [6]. Chitosan, as a biomaterial, represents the most well-known natural coagulant. Biocoagulants are mainly used to minimize the use of chemical coagulants. Chitosan powder is dissolved using acetic acid to treat wastewater. Various types of wastewater from the food industry have been treated using chitosan, such as meat [7], dairy products [7,8], bagasse [9], and egg [10], as well as various turbid natural waters [11]. It shows a removal rate between 70% and 90% for the total suspended solids content (TSS), and between 83% and 84% for the biological oxygen demand (BOD_5_), as basic forms used for measuring pollution in these cases.

In this context, this study is focused on the efficiency enhancement of the treatment plants for fish processing wastewater using biocoagulation, based on a fishery by-product (shrimp shell) in a liquid state. Using liquid chitosan minimizes the use of acetic acid and leads to more respect for the environment [12]. The effect of physicochemical parameters (initial pH, liquid chitosan dosage, turbidity, and biochemical oxygen demand) was studied in terms of the performance of the biocoagulation process using a Box–Behnken experimental design.

## 2. Materials and Methods

This study is divided into two parts. The first part concerns the preparation of liquid chitosan from shrimp shells (*Parapenaeus longirostris* [6]), while the second part is related to the modeling and optimization of the fish processing wastewater treatment with a biocoagulation process using the above mentioned chitosan-based biocoagulant.

### 2.1. Collection of Wastewater and Shrimp Exoskeleton Samples

Wastewater effluent samples were collected at a fish processing plant located in the industrial area of Anza, Agadir city, Morocco. The samples were stored at 4.0 ± 3 °C for a period of 48 h after collection, before determining the physicochemical qualities of the effluents (temperature, pH, turbidity, and biological oxygen demand over 5 days (BOD_5_)) using the ‘Standard methods for water and wastewater, APHA, 2017’ [13].

The *Parapenaeus longirostris* exoskeletons used in this study came from a local market in Agadir city in southern Morocco. The shrimp shells were repeatedly washed with water to remove the parts of the pulp and viscera stuck to the exoskeleton. The shrimp shells were dried at 220 °C for 15 min (Figure 1a), then crushed and carefully sieved to a size of 500 μm.

### 2.2. Preparation of the Natural Coagulant

The chitosan amino functionality leads to chemical reactions such as grafting, acetylation, chelation of metals, quaternization, and reactions with aldehydes and ketones, etc. to offer a variety of products with different properties, such as being non-toxic, anti-ulcer, non-allergenic, anti-acid, anti-fungal, antibacterial, andanti-viral, as well as total biodegradability and biocompatibility, etc. [14]. The antimicrobial property of chitosan is strongly affected by parameters such as pH [15]; therefore, the liquid chitosan could be conserved without alteration. Treatment of chitin using strong aqueous HCl improves its solubilization capacity. It can be used to convert the solid state of β-chitin to α-chitin [16]. The important enzymatic and chemical transformation properties of chitin are based on the molecular form (α-chitin or β-chitin). In light of numerous investigations, the α-form has been found to be less responsive than the β-form [16].

In the presence of amino groups, the pH alters the charged state and the properties of chitosan [17]. These amines are protonated at low pH and charge positively, making chitosan a cationic water-soluble polyelectrolyte [14]. However, increasing the pH above 6 causes the chitosan polymer to lose its charge, be insoluble, and deprotonate its amines [14]. The soluble–insoluble transition occurs at a pKa value between pH 6 and 6.5 [14]. Treating chitin with an alkaline solution allows obtaining chitosan with fibers having a resistance similar to those of viscose fibers [18]. To avoid solvent removal problems, the chitosan treatment was carried out with environmentally friendly reagents.

Chitosan was obtained from shrimp exoskeleton in two steps [19,20]. Chitin was extracted from shrimp shells from which chitosan were subsequently recovered. Details are as follows:

#### 2.2.1. Obtaining Chitin

Shrimp shell powder (160 g) was treated with 1 N HCl (Sigma-Aldrich, Saint-Louis, MO, USA) (1 L) for 2 h at 50 °C to remove minerals, and then filtered and washed until the pH of the filtrate was neutral. The filtered powder was dried at 200 °C for 4 h and digested further with 2.5 N NaOH for 2 h at 50 °C to remove proteins and other macromolecules, and then filtered and washed many times until a neutral pH. The chitin was dried in an oven at 80 °C for 24 h and bleached with 5% of NaClO (ACE, Pescara, Italy) 1:10 for 30 min before extracting chitosan.

#### 2.2.2. Obtaining Chitosan

The bleached chitin was treated with sodium hydroxide solution of 12 N (Sigma-Aldrich, Saint-Louis, MO, USA) in an oil bath for approximately 4 h at 140 °C, filtered, and adjusted using HCl until neutral pH. The solution was placed in an oven for 4 h at 70 °C and removed in a liquid state (Figure 1b).

The chitin/chitosan solubility problem complicates their extraction. Liquid chitosan was used without solubilization to assess its performance.

### 2.3. X-ray Diffraction (XRD)

The structure of raw shrimp shell powder was characterized using an X-ray diffractometer (Bruker D8 Twin, Jena, Germany), using monochromatic λ(KαCu) = 1.5418 Å radiation at 40 kV and 40 mA. All of the samples were tested in the range 3° < 2θ < 60°, at a step size of 0.02°, and a scan rate of 0.3 s/step.

### 2.4. Fourier Transforms Infrared Spectroscopy (FTIR)

The Fourier transform infrared (FTIR) spectra of all the produced samples were examined using attenuated total reflection Fourier transform infrared spectroscopy (ATR-FTIR, SHIMADZU, IRAffinity-1S, Paris, France). This device was equipped with a Jasco ATR PRO ONE module (Jasco, Paris, France) with 16 cm^−1^ resolution at room temperature in the wavelength spectrum from 4000 cm^−1^ to 400 cm^−1^.

### 2.5. Scanning Electron Microscopy and Energy-Dispersive X-ray Spectroscopy (SEM/EDX)

The surface morphology of raw shrimp shells was observed using a scanning electron microscope (SEM) Supra 40 Vp Gemini Zeiss Column (SEM, JEOL, JSM-IT200), with a maximum voltage of 20 kV. The elemental composition of the sample was evaluated using the associated energy dispersive X-ray analysis (EDX) coupled with SEM (JEOL, Akishima, Tokyo, Japan).

### 2.6. Jar-Test

Raw FPW (500 mL) was placed in six 1 L glass beakers to examine the dosages of chitosan and the pH required for the optimum performance. The coagulant dosages initially selected ranged from 3 to 9 mL. The dosages were further examined at various pH conditions to test their efficiency and suitability at a wider range, from 3 to 12. The test was carried out in three phases: an initial rapid mixing for 5 min, followed by a slow mixing for 20 min, and finely a final settling step for 30 min. After this, the supernatant was withdrawn for analyses. To assess the efficacy of chitosan for wastewater treatment, the following characteristics were determined: turbidity, temperature, pH, conductivity, and BOD_5_. The process efficiency was presented in terms of removal rate (R) for each parameter (turbidity and BOD_5_) and was calculated using the following equation:(1)$$\mathrm{Removal}\;\mathrm{Rate}\;\left[\%\right]=\frac{\left(Ci-Cf\right)}{Ci}\times 100\%$$
where: C*i* and C*f* are the initial and final values, respectively, for each studied parameter.

### 2.7. Analytical Methods

In order to optimize the FPW treatment using the biocoagulation process, the following three factors were taken into consideration as input variables in the experimental design model: coagulant concentration (X_1_), initial pH (X_2_), and temperature (X_3_). Table 1 shows the levels of the parameters chosen on the basis of preliminary tests. The coagulant concentration was limited to the range between 3 and 9 mg/0.5 L due to the lack of efficacy below/above these concentrations. Either deficiency or excess of positive charges added to the pollutants influence on the coagulation capacity. The pH as a parameter influencing the present process was limited to the alkaline medium, taking into consideration the biocoagulant effectiveness for this type of wastewater. However, cationic coagulants, such as chitosan, have been shown to be effective at higher temperature in the treatment of wastewater.

The experimental design, coefficient determination of the obtained models, data analysis, and 2D and 3D graphical plots were obtained using NemrodW software (Version 2000-D, LPRAI, Marseille, France) (Table 2). The experiments were performed in random order to avoid a systematic error. In addition, three central replicates were also added to the experimental design, to calculate the pure experimental error [21]. In this study, a full three-level Box–Behnken factorial design was used [22]. The response surface and the desirability function were used to define the design space and find the optimal conditions. The turbidity (Y_1_) and BOD_5_ (Y_2_) deletions were considered as the responses of the designed experiments. A total of 15 experiments were performed. In order to estimate the optimal conditions, the response variable was fitted by an empirical second-order polynomial model, established using a response surface methodology in the form of the following equation:(2)$$Y=\alpha 0\overset{k}{\underset{i=1}{\sum }}\alpha 1\mathrm{sp}+\overset{k-1}{\underset{i=1}{\sum }}\overset{K}{\underset{j=2}{\sum }}\alpha 2\mathrm{spons}+\overset{k}{\underset{i=1}{\sum }}\alpha 1{\mathrm{sXi}}^{2}+\varepsilon$$
where α0, αi, αii, and αij are the regression coefficients for the intercept, linear, quadratic, and interaction terms, respectively; Xi and Xj are the independent variables (experimental parameters); and ε is the error [23].

## 3. Results and Discussion

The initial parameters of the raw wastewater used in this study are provided in Table 3. In accordance with Moroccan environmental legislation, the presented values for all water quality indicators, except the pH, exceeded the discharge limits into natural effluents [24].

### 3.1. Shrimp Shell Characterization

#### 3.1.1. Analysis by Scanning Electron Microscope

The different mixtures of raw shrimp shells were observed to identify the elemental composition of the material (Figure 2 and Figure 3). Various structures visualized by SEM were analyzed using EDX-SEM images. Figure 2a–c shows that the dried biomaterial was not homogeneous. The surface of Figure 2a,b has a relatively smooth and, therefore, less porous surface. Figure 2c shows a microporous and rough surface structure with different sizes and morphologies of grain.

The presence of holes on the specific surface corresponds to the pores present in the third image (Figure 2c). The morphology of this biomaterial can facilitate the adsorption of anions and metallic elements, due to the irregular surface [25].

#### 3.1.2. Scanning Electron Microscopy and Energy-Dispersive X-ray Spectroscopy (SEM/EDX)

As seen in Figure 3, the EDX analysis for Figure 2b with a smooth surface and Figure 2c with a rough surface show similar elements. Only C, O, and Ca signals could be observed; these are known as the main components of raw shrimp shells. The cations, Na, Mg, P, Si, Cl, K, and N were found in very small quantities.

Chitin exists in raw shrimp shells in two different abundant polymorphic forms (α and β). The chitin chains in both structures are presented as sheets. Several intra-sheet hydrogen bonds tightly retain chitin sheets, with α and β chains packed in antiparallel arrangements [26,27,28]. Therefore, a tight network is formed, dominated by rather strong C-O-NH hydrogen bonds. A distance of about 0.47 nm is maintained in these chains [29]. Such a characteristic is not found in the structure of β-chitin, which is therefore more sensitive than α-chitin to intracrystalline swelling [29,30,31,32]. The current model of the crystal structure of α-chitin indicates that the inter-sheet hydrogen bonds are divided into two sets, with half-occupation in each set [29]. These aspects highlight the insolubility and intractability of chitin.

We can conclude that calcium carbonate CaCO_3_ is the main element of shrimp shells, according to the EDX analysis and many similar studies [33,34,35,36].

#### 3.1.3. Analysis by Infrared Spectroscopy

The infrared spectrum of the raw shrimp shells (Figure 4) shows the absorption bands found in the spectrum IR of the biomolecules.

Characteristics peaks of calcite (871 cm^−1^, 1404 cm^−1^, and 2522 cm^−1^) were found in the IR, in addition to characteristic peaks of α-chitin (1658 cm^−1^, 1319 cm^−1^, 1072 cm^−1^, 586 cm^−1^) (Figure 5) [37,38,39,40,41,42].

The IR confirms the presence of peaks characteristic of (OH) hydroxyl groups and (NH) primary amine, represented by a broadband at 3400 cm^−1^, which may be an attribution to the hydrogen bonds. C-H stretching vibration is also presented by two spectra: 2900 cm^−1^ and 1395 cm^−1^. A strong band at 2400 cm^−1^ appears in the experimental spectrum, due to the vibration band within CO_2_ molecules, as stated by [41,42].

The differences with the shrimp exoskeleton IR spectra in other studies may be attributed to the species of shrimp used.

#### 3.1.4. Analysis by XRD Spectroscopy

The X-ray diffraction pattern for raw shrimp shells is given in Figure 5. The two mean peaks at 20° and 30° correspond to the diffraction planes (411) and (435), respectively. The latter peak is stronger than the first. The patterns show the presence of carbon and oxygen as the major constituents of proteins and calcium as the principal element of calcite [43,44].

According to Figure 6, the majority of the peaks shown in the X-ray diffraction calcite matched the peaks of the raw shrimp shells studied. The mean peak of the diffraction revealed the nature of the material. That this biomaterial shows great compatibility, confirms the significant presence of calcite in the sample analyzed. Similar to many studies, carbonates were mostly in the form of calcium bicarbonates, as has been reported for crustacean shells [42,45,46].

### 3.2. Effect of Coagulant Dosage

The coagulant dose is an important parameter in coagulation [47,48]. The optimum dosage determines the performance of the coagulants, in terms of viability and economy [49].

A lower or higher dosage than the optimum may result in insufficient coagulation and lead to lowering the effect, due to the stabilization of charges, and will not improve the efficiency further [50]. The effect of coagulant doses (from 3 to 9 mL/0.5 L) at a 30 min sedimentation time on the chosen parameters (turbidity and BOD_5_) with the alkaline media of the wastewater is shown in Figure 7. It can be noted that a 3 to 9 mL dosage worked well at alkaline pH (pH = 10), in terms of turbidity, whereas for BOD_5_ it required a 6 mL dose. The coagulating efficiency of chitosan was clear, without dissolving it in acetic acid. The nature of the treated water (fish processing wastewater) made the liquid chitosan function in the presence of sodium hydroxide, which favored the alkaline media. The chitosan particles, in combination with the dissolved sodium hydroxide particles, coagulated the pollutants presented in the FPW. The lack of efficacy of a dose greater than 6 mL may have been due to the destabilization of the charges, which leads to impaired coagulation. Based on these experiments, it was concluded that the 6 mL chitosan dosage was efficient for reducing turbidity (98.32%) and BOD_5_ (53.5%).

Commercial chitosan has been used to treat different types of wastewater. Based on previous studies, using 1.5 g/L of commercial chitosan to treat tanning industry wastewater reduced total suspended solids and biological oxygen demand by 89% and 33.3%, respectively [51]. It has also been tested on textile wastewater, with a reduction rate of 90%, in terms of turbidity reduction [50]. Another study found that treating olive oil wastewater with 400 mg/L chitosan at pH 4.3 removed 81% of the total suspended solids [52].

### 3.3. Effect of pH

The effect of pH on the turbidity removal was tested at different pH values, from 2 to 12. Figure 8 shows the effect of pH on liquid chitosan as a coagulant for turbidity removal with 30 min of sedimentation. It can be noted that chitosan was sensitive to pH changes and required near alkaline pH conditions for effective turbidity and BOD_5_ removal. A pH value of 10 was found to be ideal for a higher performance of chitosan for FPW. The sodium hydroxide presented in the liquid chitosan solution also had some effect, as a particle which increases the pH of the solution; at the same time, a large number of sodium ions dissolved in the coagulating solution as electrolytes will promote coagulation by destabilizing the system [53]. A removal efficiency of around 98% was noted in an alkaline medium (pH = 10). Lower removal efficiencies were observed at neutral pH, where BOD_5_ showed a percentage removal of 53%.

A similar observation reported that chitosan as a coagulant has a good performance in alkaline media [52,54,55]. Chitosan powder as a coagulant showed a maximum removal of 26–28% of total solids at pH 5.5. What emerged clearly from our study is that pH 6 has a turbidity removal percentage of 15% [56]. Another study found that chitosan powder showed an effective removal of various pollutants from aquaculture wastewater through coagulation, adsorption, or disinfection [57]. Therefore, the type and quality of wastewater affects the variation in pH and, hence, the effectiveness of the treatment [58].

### 3.4. Optimization Using a Box–Behnken Experimental Design

#### 3.4.1. Statistical Analysis

Preliminary experiments were carried out to study the effect of the most significant parameters on the effectiveness of the FPW treatment. The relationship between the three variables (coagulant dose, initial pH, and temperature) and the two important process responses (turbidity and BOD_5_ removal efficiency) for the biocoagulation process was analyzed using the surface of modeling response. In addition, the optimization of the variables studied on the basis of the experiments was carried out using the software NemrodW. The statistical analyses employed were Student’s and ‘t’ test (Table 4).

Analysis of variance (Table 5) for turbidity and BOD_5_ removal shows that the fitted second-order response surface model was highly significant, with a significance value of Turbidity Signif = 0.0663 *** and BOD_5_ Signif ≤ 0.01 ***.

Student’s ‘t’ test was used to determine the significance of the regression coefficients of the variables. The significance of the test reaching a value less than 0.01 indicates that the test parameter is significant at a significance level of 1% (Table 4). Compared to other studies adopting a RSM and Box–Behnken statistical analysis for the optimization of industrial wastewater treatment [22,59,60], our models were considered highly significant and indicated excellent correlations between the experimental results and predicted values of BOD and turbidity removal uptake with these models.

#### 3.4.2. Modeling Approach of the Response Surface to Optimize the Variables Studied

Box–Behnken statistical analysis was used to determine the single and combined effects of independent variables on the responses and outcomes, as well as on experimental conditions. According to the sum of the squares of the sequential model, the models were selected based on the highest order polynomials, where the additional terms were significant. The experimental results were evaluated and the approximation function between the FPW (Y) treatment and the variables was studied on the basis of the estimated parameters (Table 4) for turbidity and BOD_5_ (1)–(2), respectively:(3)$$Y_{1}=98.320-1.633{\;X}_{1}-1.481{\;X}_{2}-2.069{\;X}_{3}-4.424{\;X}_{1}^{2}-0.631{\;X}_{2}^{2}-2.021{\;X}_{3}^{2}+0.765{\;X}_{1}X_{2}+1.165{\;X}_{1}X_{3}\;-0.992{\;X}_{2}X_{3}$$
(4)$$Y_{2}=53.500-7.358X_{1}+5.920X_{2}+3.020X_{3}-14.191X_{1}^{2}-1.336X_{2}^{2}-11.021X_{3}^{2}-0.512X_{1}X_{2}-1.402X_{1}X_{3}\;-3.832X_{2}X_{3}$$

#### 3.4.3. Response Surface Plotting and Optimization of Turbidity Removal

In order to provide a better explanation of the effects of the independent variables and their interactions, two- (2D) and three-dimensional (3D) response surface plots were drawn as a function of three parameters at a time, holding the third parameter as fixed. The following graphical representations (Figure 9, Figure 10 and Figure 11) show the behavior of turbidity under the simultaneous change of the three variables, while fixing the third one.

As shown by the contour lines corresponding to the removal of turbidity, the shape of a dome revealed the strong interaction between the tested parameters; the depression in the response areas/surfaces indicated that the optimum conditions for maximum turbidity removal were exactly within the design limit. Thus, at a fixed pH of 12, maximum turbidity removal percentages were obtained when the dose of coagulant was 5.5 mL/500 mL and while the temperature was between 20 °C and 24 °C. Turbidity removal decreased when the pH of the wastewater was in an acid media, which could be explained by the fact that chitosan requires sufficient alkalinity to be hydrolyzed. From Figure 9, it can be deduced that the maximum elimination of turbidity was observed when the dose of coagulant was fixed at 5.5 mL, under a fixed temperature at 20 °C, and when the pH varied from 10 to 12. The contour curves of the pH as a function of temperature at the dose of coagulant maintained at 5.5 mL presented in Figure 10 show that the turbidity decreases with the increase in pH, whatever the temperature. The highest turbidity removal was obtained when the pH values were above 8. Figure 11 shows the interaction between temperature and the coagulation dose, when the pH was set at 10.5. The temperature increases when the coagulant dose concentration decreases.

#### 3.4.4. Response Surface Plotting and Optimization of BOD_5_ Removal

The response Equation (2) was used to visualize the effects of the experimental parameters on the responses under optimized conditions, using two-dimensional (b) and three-dimensional plots (a). Figure 12, Figure 13 and Figure 14 show response surface plots of the biocoagulation of FPW as a function of two parameters, while the third was held at a constant level.

Figure 12 shows that as the pH increases, the removal of BOD_5_ increases independently of the dose of coagulant at a fixed temperature of 20 °C. When the dose of coagulant is set at 5.5 mL, the interaction between temperature and pH is reciprocal (Figure 13). The pH increased with temperature to an optimum condition. In Figure 13, the percentage removal of BOD_5_ increased with pH and coagulant dose over a range of 5–7 mL. An approximately 90% BOD_5_ removal was obtained when the dose of coagulant was set around 5.5 mL.

The 2D and 3D surface plots show that the maximum turbidity and the elimination of BOD_5_ from FPW were obtained under alkaline media conditions (higher than 10.5), for a dose of coagulant of 5.5 mL, and at a temperature between 20 and 24 °C. The optimal experimental conditions for turbidity and BOD_5_ removal rates were obtained by analyzing the response surface contour plots (graphical analysis). However, these values were determined from a mathematical point of view, using a desirability function approach (Desirability/Monte-Carlo) [61], which gives the optimal results summarized in Table 6, for a desirability function value (D) of 0.94168.

For these established optimal conditions, the theoretical values for Y_1_ and Y_2_ were experimentally verified and repeated more than three times (confirmation run), obtaining a good concordance, confirmed by the close values: Y_1_exp = 98% and Y_2_exp = 53%, respectively.

In this study case, liquid chitosan under the optimal conditions changed the dark brown color of raw FPW (Figure 15a) to significant discoloration, compared to distilled water (Figure 15c). During the biocoagulation treatment, large flocks appeared before the sedimentation phase. Thirty minutes of sedimentation phase was sufficient to remove the treated FPW without any flocculant addition.

Similar observations have been reported for large chitosan flocks in other studies. Lichtfouse and collaborators [62] reported the formation of large flocks after direct biocoagulation of wastewater. Ruhsing and collaborators [63] also reported that wastewater treated with chitosan produced the largest flocks and that the settling rate was about 1.5 times faster than that of poly-aluminum chloride.

### 3.5. Comparison with Other Coagulants

Liquid chitosan was compared to ferric chloride and commercial chitosan. The preparation of the two coagulants was made according to the optimal conditions for each one according to the literature. For ferric chloride, the optimal pH for fish processing industries is pH = 6. A dose optimization was carried out to define the optimal dose for this effluent (5.5 mL).

Commercial chitosan was studied in a wide pH range, from 1 to 12. The optimum pH was found to be 11. Doses were also studied in the optimum pH (9 mL).

Table 7 shows a comparative experimental study for three coagulants.

As shown in Table 7, liquid chitosan has a good cleaning ability for fish processing wastewater. In terms of turbidity, there was no significant difference between the three coagulants, while liquid chitosan and commercial chitosan showed a good reduction of the BOD_5_ parameter. All coagulants tested were beyond the Moroccan standard direct discharge limit. In this study, the coagulants were tested without any addition of flocculant. Therefore, the results of Table 7 are very encouraging for further study of this biocoagulant, in order to reduce the use of chemical in terms of chemical coagulants or dilution of chitosan with acetic acid (non-environmental product).

## 4. Conclusions

A novel liquid chitosan based biocoagulant for treatment of a fish processing wastewater from a Moroccan plant was successfully prepared from the most abundant fish by-product, namely shrimp shells (*Parapenaeus longirostris*), which was characterized using different techniques: XRD, FTIR, SEM/EDX. The use of liquid chitosan prepared from the fish solid waste will minimize the acetic acid impact on the environment and represents, at the same time, a promising solution for solid waste management. Chitosan, being a natural coagulant, respects the ‘bio concept’ and could be successfully used to minimize the use of chemical coagulants, being effective in the removal rate of turbidity (98%) and BOD_5_ (53%) from fish processing wastewater. These results were obtained for the three most significant experimental parameters (coagulant dose, initial pH, and temperature) of the wastewater treatment, by applying a response surface methodology using a Box–Behnken experimental design and a desirability function approach (desirability/Monte-Carlo). Therefore, the optimal experimental conditions correspond to an alkaline media (pH = 10.5), a dose of 5.5 mL biocoagulantin, 0.5 L of fish processing wastewater, and a temperature of 20 °C.

The obtained results showed a good concordance between the experimental values and the values calculated by the prediction models, which were experimentally verified by additional confirmation run. The model validation demonstrated that the response surface methodology approach was appropriate for optimizing the biocoagulation process, in terms of the efficiency enhancement of the removal rate of turbidity and BOD_5_. For both of these regression equations were obtained, which could be used as models for the simulation and control of industrial wastewater treatments performed using a biocoagulation process.

## Figures and Tables

**Figure 1 materials-14-07133-f001:**
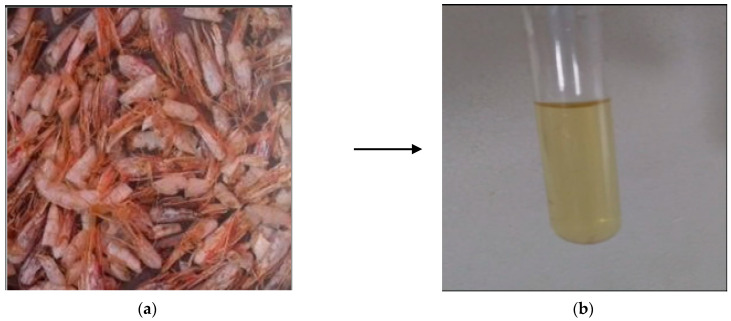
Transformation of shrimp shells to liquid chitosan: (**a**) dried shrimp exoskeleton; (**b**) liquid chitosan.

**Figure 2 materials-14-07133-f002:**
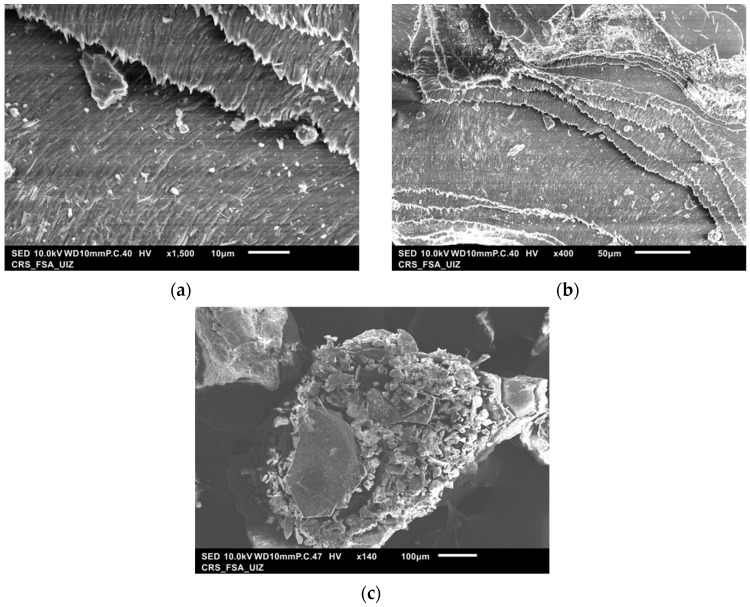
Scanning electron microscope image (SEM) of the raw shrimp shells (**a**): 10 μm, (**b**): 50 μm, (**c**): 100 μm.

**Figure 3 materials-14-07133-f003:**
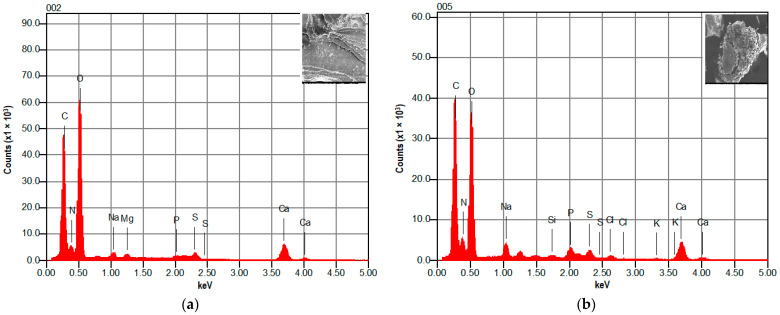
EDX spectrum of raw shrimp shell: (**a**) for 10 μm SEM image and (**b**): for 50 μm SEM image.

**Figure 4 materials-14-07133-f004:**
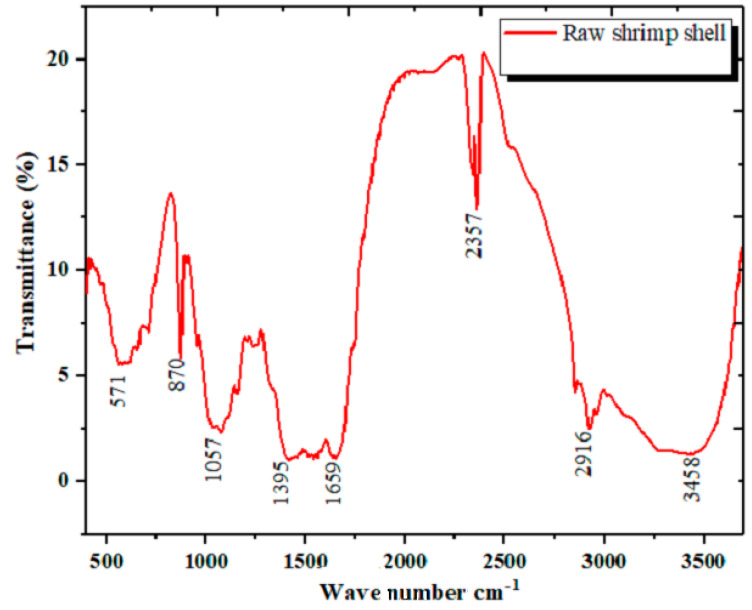
Infrared spectrum of raw shrimp shell.

**Figure 5 materials-14-07133-f005:**
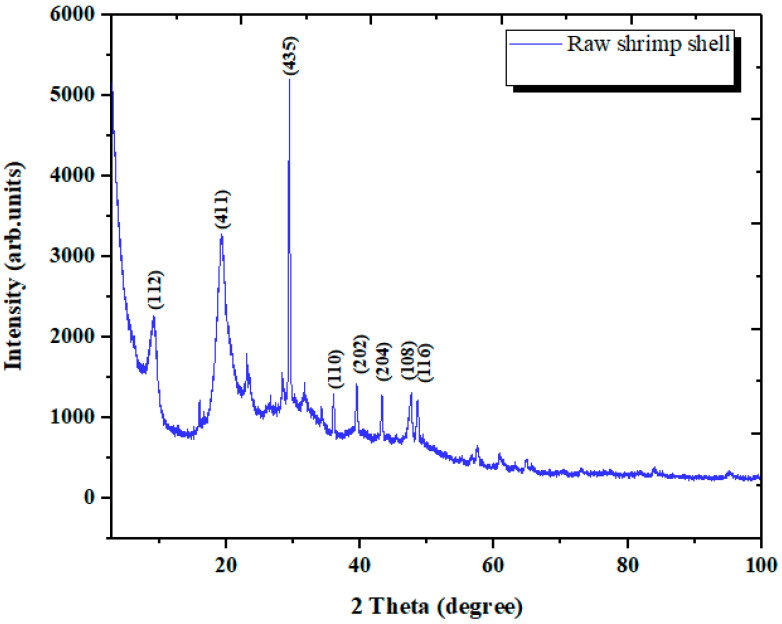
XRD spectrum of raw shrimp shell.

**Figure 6 materials-14-07133-f006:**
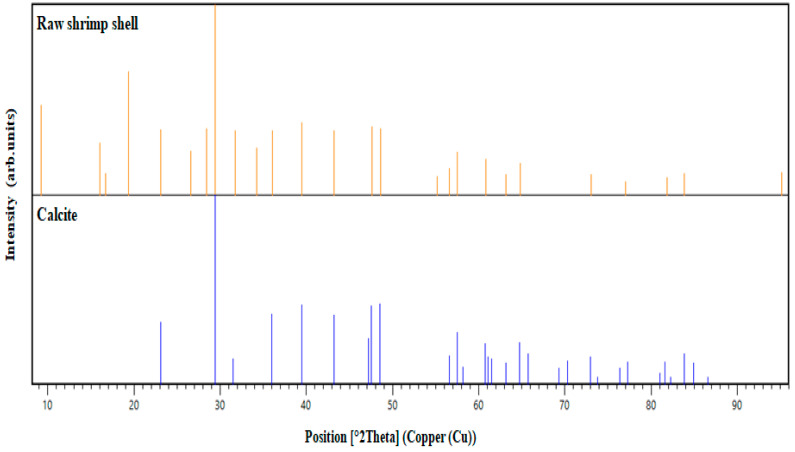
Comparison of XRD patterns of raw shrimp shell and calcite.

**Figure 7 materials-14-07133-f007:**
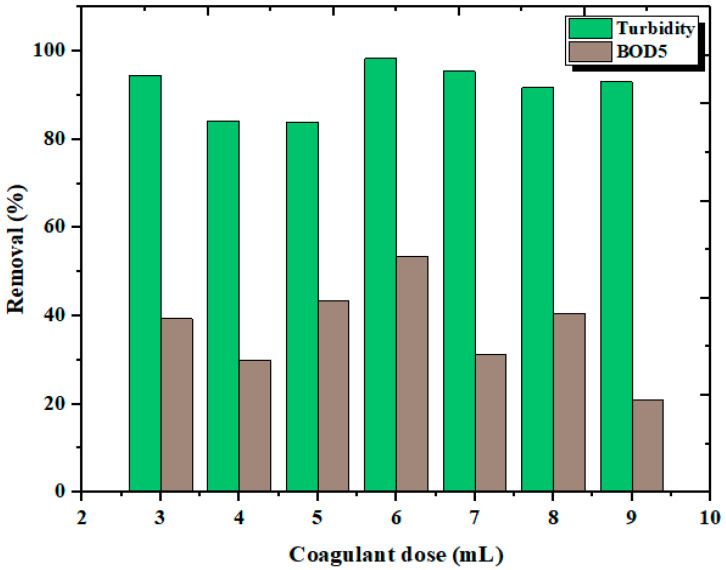
Effect of liquid chitosan doses on turbidity and BOD_5_ removal.

**Figure 8 materials-14-07133-f008:**
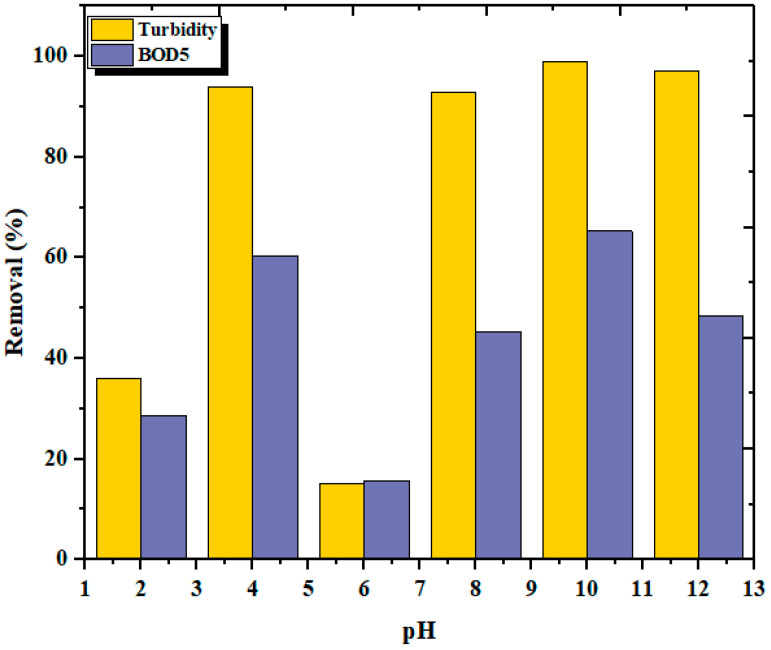
Effect of pH on the turbidity and BOD_5_ removal rates.

**Figure 9 materials-14-07133-f009:**
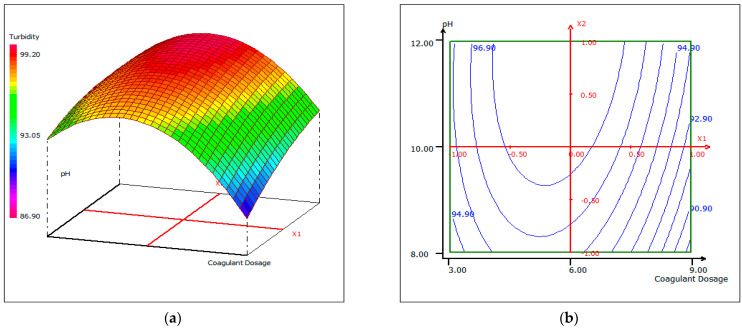
Three-dimensional (**a**) and two-dimensional (**b**) response surface plots for the turbidity of the treated fish processing wastewater using a biocoagulation process as a function of coagulant dose and pH, when the temperature was maintained at 20 °C.

**Figure 10 materials-14-07133-f010:**
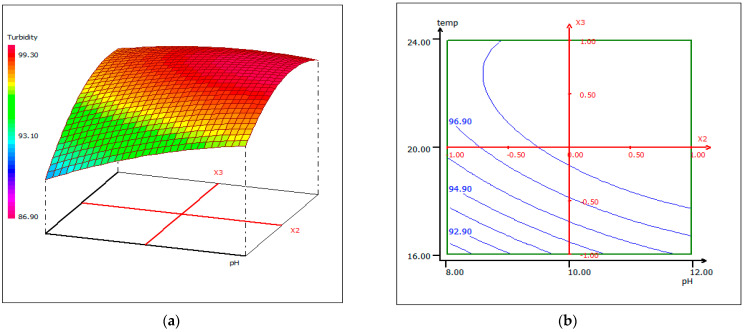
Three-dimensional (**a**) and two-dimensional (**b**) response surface plots for the turbidity of the treated fish processing wastewater using a biocoagulation process as a function of temperature and pH, when the coagulant dose was maintained at 5.5 mL.

**Figure 11 materials-14-07133-f011:**
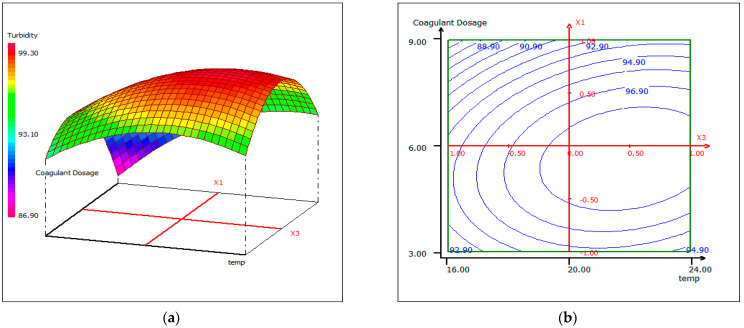
Three-dimensional (**a**) and two-dimensional (**b**) response surface plots for the turbidity of the treated fish processing wastewater using a biocoagulation process as a function of coagulant dose and temperature, when the pH was maintained at 10.5.

**Figure 12 materials-14-07133-f012:**
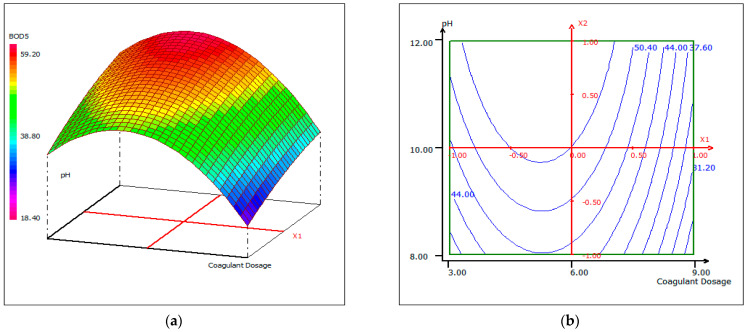
Three-dimensional (**a**) and two-dimensional (**b**) response surface plots for the BOD_5_ of the treated fish processing wastewater using a biocoagulation process as a function of coagulant dose and pH, when the temperature was maintained at 20 °C.

**Figure 13 materials-14-07133-f013:**
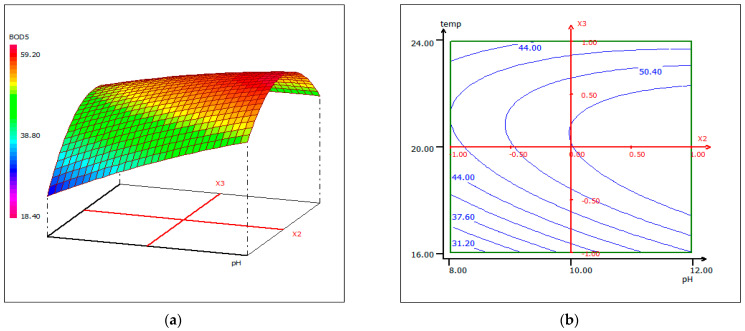
Three-dimensional (**a**) and two-dimensional (**b**) response surface plots for the BOD_5_ of the treated fish processing wastewater using a biocoagulation process as a function of temperature and pH, when the coagulant dose was maintained around 5.5 mL.

**Figure 14 materials-14-07133-f014:**
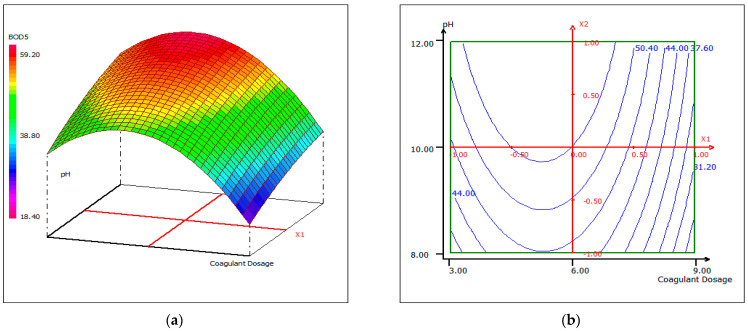
Three-dimensional (**a**) and two-dimensional (**b**) response surface plots for the BOD_5_ of the treated fish processing wastewater using a biocoagulation process as a function of coagulant dose and temperature, when the pH was maintained at 10.5.

**Figure 15 materials-14-07133-f015:**
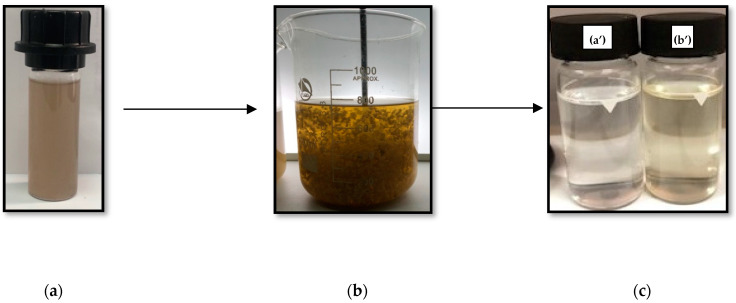
Representation of the discoloration process after treatment with liquid chitosan: (**a**) raw FPW; (**b**) FPW during the biocoagulation process; (**c**) comparison of FPW after biocoagulation using liquid chitosan (**b’**) with distilled water (**a’**).

**Table 1 materials-14-07133-t001:** The range and levels of experimental variables (parameters).

Coded Variables	Parameters	Coded Level		
		−1	0	+1
X_1_	coagulant concentration (mg/0.5 L)	3	6	9
X_2_	initial pH	8	10	12
X_3_	temperature (°C)	16	20	24

**Table 2 materials-14-07133-t002:** Experimental design matrix for FPW treatment through biocoagulation.

Run No.	Experimental Parameters			Responses	
	X_1_ Coagulant Dose (mL)	X_2_ Initial pH	X_3_ Temperature (°C)	Y_1_ Turbidity Removal (%)	Y_2_ BOD_5_ Removal (%)
1	3.00	8.00	20.00	93.40	39.52
2	9.00	8.00	20.00	89.10	26.74
3	3.00	12.00	20.00	95.90	50.23
4	9.00	12.00	20.00	94.66	35.40
5	3.00	10.00	16.00	92.97	31.21
6	9.00	10.00	16.00	86.88	18.39
7	3.00	10.00	24.00	94.54	39.33
8	9.00	10.00	24.00	93.11	20.90
9	6.00	8.00	16.00	91.54	26.10
10	6.00	12.00	16.00	95.42	47.76
11	6.00	8.00	24.00	97.90	40.53
12	6.00	12.00	24.00	97.81	46.86
13	6.00	10.00	20.00	98.30	53.39
14	6.00	10.00	20.00	98.32	53.50
15	6.00	10.00	20.00	98.31	53.61

**Table 3 materials-14-07133-t003:** Physicochemical characterization of FPW.

Parameter	Value			Moroccan Standard Limits for Direct Discharge into Effluents [24]
	Min	Mean	Max	
temperature	12	18.5	25	30 °C
pH	6.62	7.06	7.50	5.5–9.5
conductivity	9000	15,500	22,000	2700 µS/cm
turbidity	850	925	>1000	-
BOD_5_	1700	2345.5	2990.9	100 mg O_2_/L

**Table 4 materials-14-07133-t004:** Statistical analysis of the regression coefficients estimated due to the FPW variation.

	**Parameter**		**Standard Deviation**		**R_2_**		**R_2_ Adjusted**		**R_2_ Predicted**	
	Turbidity		0.759		0.983		0.953		0.730	
	BOD_5_		1.549		0.994		0.984		0.906	
**Coefficient**	**Value**		**F. Inflation**		**Standard Deviation**		**T Exp**		**Significance (%)**	
	**Turbidity**	**BOD_5_**	**Turbidity**	**BOD_5_**	**Turbidity**	**BOD_5_**	**Turbidity**	**BOD_5_**	**Turbidity**	**BOD_5_**
b_0_	98.32	53.50	-	-	0.44	0.90	224.31	59.83	<0.01 ***	<0.01 ***
b_1_	−1.63	−7.36	1.00	1.00	0.27	0.55	−6.08	−13.44	0.174 **	<0.01 ***
b_2_	1.48	5.92	1.00	1.00	0.27	0.55	5.52	10.81	0.268 **	0.0118 ***
b_3_	2.06	3.02	1.00	1.00	0.27	0.55	7.71	5.51	0.0587 ***	0.268 **
b_1-1_	−4.42	−14.19	1.01	1.01	0.40	0.81	−11.20	−17.61	<0.01 ***	<0.01 ***
b_2-2_	−0.63	−1.34	1.01	1.01	0.40	0.81	−1.60	−1.66	17.1	15.8
b_3-3_	−2.02	−11.85	1.01	1.01	0.40	0.81	−5.12	−14.70	0.372 **	<0.01 ***
b_1-2_	0.77	−0.51	1.00	1.00	0.38	0.77	2.02	−0.66	10.0	53.7
b_1-3_	1.17	−1.40	1.00	1.00	0.38	0.77	3.07	−1.81	2.78 *	13.0
b_2-3_	−0.99	−3.83	1.00	1.00	0.38	0.77	−2.61	−4.95	4.74 *	0.429 **

*** Extremely significant; ** Very significant; * Significant.

**Table 5 materials-14-07133-t005:** Analysis of variance (NemrodW) for the models of the biocoagulation process for FPW treatment.

Sources of Variation	Sum of Squares		Degree of Freedom		Mean Square		Ratio		Significance (%)	
	Turbidity	BOD_5_ Removal	Turbidity	BOD_5_ Removal	Turbidity	BOD_5_ Removal	Turbidity	BOD_5_ Removal	Turbidity	BOD_5_ Removal
regression	167.68	2.02869 × 10^3^	9	9	18.63	2.25410 × 10^2^	32.32	93.95	0.07 ***	<0.01 ***
residual	2.88	1.19955 × 10^1^	5	5	0.58	2.39910				
total	170.56	2.04069 × 10^3^	14	14						

*** Extremely significant.

**Table 6 materials-14-07133-t006:** The optimal experimental conditions calculated using a desirability function approach (Desirability/Monte-Carlo), and the theoretical values for Y_1_ and Y_2_.

X_1_ Coagulant Concentration (mg/0.5 L) Coded Value Real Value	X_2_ Initial pH Coded Value Real Value	X_3_ Temperature (°C) Coded Value Real Value	Y_1_ Turbidity Removal Rate (%)	Y_2_ BOD_5_ Removal Rate (%)
−0.23982	0.22863	−0.20584	98.488	54.783
5.2805	10.457	19.177		

**Table 7 materials-14-07133-t007:** Comparative experimental results for three coagulants used for FPW treatment.

Parameter	Value before Treatment	Value after Treatment			Moroccan Standard Limit of Direct Discharge [24]
		Liquid Chitosan	Ferric Chloride	Commercial Chitosan	
turbidity	>1000	10.7	15.6	13.8	-
BOD_5_	2990.9	201	412.5	209	100 mg O_2_/L

## Data Availability

Data is contained within the article.

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
