# Peer review of "Novel Liquid Chitosan-Based Biocoagulant for Treatment Optimization of Fish Processing Wastewater from a Moroccan Plant"

_materials, 2021, doi:10.3390/ma14237133_

Round 1

Reviewer 1 Report

This is an interesting manuscript reporting the optimization of treatment of fish processing wastewater using liquid chitosan biocoagulant. I will recommend its publication after minor revision.

General comments: The language of the manuscript requires special attention.

Specific comments:

Line 82: Please specify the meaning of “5” in “BOD5”.

Line 85: Specify other factors rather than “etc”.

Line 106: Give rational for the size selection of “500um”

Line 108-110: Language requires attention.

Line 112-117: Cite references.
Line 127: Specify city, state/country.

Line 132: Specify model.

Section 2.7: It is essential to give the rational for choosing the levels of the variables listed in Table 1.

Table 4 & 5: Significance (%) values are not specified. This reviewer feels that the values are F values. It is important to specify the significance (%) at the Table footnotes (i.e., the meanings of *, ** and ***).

Line383-385: It is essential to specify here how many times the experiments were repeated for the experimental validation. Also statistical comparison tests are required to calculate the significance between the model and experimental values of Y1 and Y2.

Author Response

Dear Reviewer 1,

We appreciate the reviewer’s efforts and their time for carefully reading, editing, and commenting on our manuscript. We are very thankful for their critical comments and constructive suggestions, which helped us to improve the manuscript quality. We have revised our manuscript based on the reviewer’s comments and gave detailed answers/justifications to their comments/suggestions/queries. The responses to the reviewer’s comments are given below using red color type.

All the changes done in the manuscript during this revision are shown directly in the revised text highlighted in yellow color.

Reviewer 2 Report

I have read up to Section 3.3 of a manuscript entitles “A new liquid chitosan based biocoagulant for treatment optimization of a fish processing wastewater from a Moroccan plant.  While the experimental design is excellent, it is a very difficult to understand the text.  Grammar and sentence structure limit accessibility to the important message being delivered in the manuscript:

  • Past tense is mixed with present tense in the same sentence;
  • Long sentences that convey little meanings;
  • Consistency lacking, e.g. Figure X and figure X, and Table Y and table Y are used interchangeably, so is hour and h.

Here are examples of my concerns:

TITLE

The “old” liquid chitosan is not mentioned anywhere and should be mentioned somewhere in the text if this is a “new” liquid chitosan. An explanation about the need for the new one should accompany the mention.  If not, is suggest the phrase “new” be delete from the title, from the first line of the Abstract and in the text.

ABSTRACT

Lines 10-13: Break this long sentence into two, perhaps something like this:

“A liquid chitosan-based biocoagulant for treating wastewater from a Moroccan fish processing plant was successfully prepared from shrimp shells (Parapenaeus-longirostris), the most abundant fish by-products in the country. The shells were previously characterized using X-ray diffraction, Fourier transforms infrared spectroscopy, scanning electron microscopy, and energy-dispersive X-ray spectroscopy.”

Lines 14-16: Rewrite the following sentence; it has no meaning:

“The using in wastewater treatment of liquid chitosan, without the addition of acetic acid, will minimize their impact to the environment and represents in the same time a promising solution for solid fish waste management.”

Lines 16-18: Rewrite the following sentence:

“In order to test the treatment efficiency of the developed biocoagulant, prepared according to the present methodology, for the industrial fish wastewater, a qualitative characterization of these effluents was previously carried out.”

Line 18-23: Break this sentence up into manageable sizes for readability:

“In order to enhance the biocoagulation efficiency in terms of removal rate of turbidity and BOD5 from the fish processing wastewater industry, the process optimization was conducted in two steps: jar-test experiments for a preliminary assessment to identify the most influential factors and the second step was to study the effects of three significant operational parameters (experimental conditions) and their interactions using the Box-Behnken experimental design.”

Lines 25-29:  Consider perhaps rephrasing these two sentences as follows:

“Regression models and response surface contour plots reveal that chitosan as a liquid biocoagulant is effective in removing turbidity (98%) and BOD5 (53%) during the treatment. The optimal experimental conditions were found to be an alkaline media (pH=10.5) and a dose of 5.5 mL biocoagulant in 0.5 L of fish processing wastewater maintained at 20 °C.”

Between Lines 31 & Line 32: Increase the font size of the image on page 2.

INTRODUCTION

Lines 35-43: Suggested re-write for this paragraph if it does not alter meanings:

“Wastewater effluents from fish processing plants (fish processing wastewater – FPW) presents an environment problem because of complication in their treatment due to different components presented in the effluents.  In Morocco, the subject of this study, the effluent volume is not only increasing, but fish processing industry is also responsible for a large quantity of fish-related solid wastes, including skins, bones, fins, scales, and swim 40 bladders, constituting 36% (and may represent up to 60%) of the raw material mass [2,3]. These solids waste are in themselves potential sources of income if enough valuation of the byproducts, the industrial wastewater is integrated into fish processing or in the effluents,”

Line 39: Check this value: 207 407 800,00. Do you mean USD 207,407,800.00?

MATERIALS AND METHODS

Lines 71-75: Rephrasing strongly is suggested here. Perhaps the following may help:

“This study is divided into two parts. The first part concerns the preparation of liquid chitosan from shrimp shells (Parapenaeus longirostris) [6]. The second part concerns the treatment of wastewater out of fish processing using the chitosan biocoagulant by the Box-Behnken response surface methodology”

Line 76: Consider renaming sub-section 2.1. as follows:

"2.1. Collection of wastewater and shrimp exoskeleton samples.”

Lines 77-82: Suggested rewrite:

“Wastewater effluent samples were collected at a fish processing plant located in the industrial area of Anza, Agadir city, Morocco. The samples were stored at 4.0 ± 3°C for a period of 48 h after collection before determining the physicochemical qualities of the effluents (temperature, pH, turbidity, and biological oxygen demand (BOD5)) by the “Standard methods for water and wastewater, APHA, 2017” [13].

Line 102-106: The paragraph is out of place here and should be moved to the renamed sub-section 2.1.

Lines 107-110: Suggested rewrite for clarity:

"Chitosan was obtained from shrimp exoskeleton in two steps [19,20]. Chitin was extracted from shrimp shells from which chitosan was then recovered. The details are as follows."

Lines 114-117: Suggested rewrite:

Shrimp shell powder (160 g) was treated with 1 N HCl (1 L) for 2 h at 50 ° C to remove minerals, filtered and washed until the pH of the filtrate was neutral. The filtered powder was dried at 200 °C for 4 h and digested further with 2.5 N NaOH for 2 h at 50 °C to remove proteins and other macromolecules, filtered and washed many times until the pH of the filtrate was. The chitin was drying in the oven at 80 °C for 24 h and bleached with 5% of NaClO 1:10 for 30 min before extracting chitosan.

Lines 119-123: This is a confusing statement. On the one hand, a solid is being washed until the pH of the flow through water is neutral. And then, (i) the same solid is now in solution, and (ii) an aqueous solution in Fig. 2b is dried at 70 °C?

The authors further make this statement: “Liquid chitosan is used without solubilization.” How do you solubilize chitosan if it is already in an aqueous phase (Fig. b)? Please, clarify.  Perhaps the following might give the authors a start:

“The chitin was treated with 12 N NaOH on an oil bath for approximately 4 h at 140 °C, filtered and washed until the pH of the flow-through water was neutral pH. The solution was dried in an oven for 4 h at 70 °C and kept in liquid state (Figure 1b). The chitin/chitosan solubility problem complicates their extraction. Liquid chitosan is used without solubilization to evaluate to assess its performance.”

Lines 141-151: What is “from 3 to 12”, is it pH or mL? Consider the following rewrite:

“Raw FPW (500 mL) was placed in six 1-L glass beakers to examine the dosages of chitosan and the pH required for optimum performance. The coagulant dosages initially selected ranged from 3 to 9 mL. The dosages were further examined at various pH conditions to test their efficiency and suitability at a wider range from 3 to 12. The test was carried out in three phases: an initial rapid mixing for 5 min, followed by a slow mixing for 20 min, and finely a final settling step for 30 minutes. After this, the supernatant was withdrawn for analyses. To assess the efficacy of chitosan on wastewater treatment, the following characteristics are determined: turbidity, temperature, pH, conductivity and BOD5. The process efficiency was presented in terms of Removal Rate (R) for turbidity and BOD5 and was calculated using the following equation:

RESULTS AND DISCUSSION

Figure captions are not adequately informative (see this recent paper in Materials for example https://www.mdpi.com/1996-1944/13/9/2178/htm). An average reader may want to know, for instance, what the numbers in parentheses ((112) for example) mean in Figure 5, or how the pattern in Figure 6 is consistent with the presence of calcite. What is the label on the response axis in Figure 6?

The authors could explain how TSS, BOD5 and turbidity tie in specifically with their interesting findings in Figures 7 & 8.

Lines 177-179: Consider this rewrite:

The initial parameter of raw wastewater used in this study are provided in Table 3. Except for pH, the values were all above Moroccan’s wastewater discharge limits in the environment [24].

Lines 259-260:  This sentence needs a rewrite for clarity.

Author Response

Dear Reviewer 2,

We appreciate the reviewer’s efforts and their time for carefully reading, editing, and commenting on our manuscript. We are very thankful for their critical comments and constructive suggestions, which helped us to improve the manuscript quality. We have revised our manuscript based on the reviewer’s comments and gave detailed answers/justifications to their comments/suggestions/queries. The responses to the reviewer’s comments are given below using red color type.

All the changes done in the manuscript during this revision are shown directly in the revised text highlighted in yellow color.
